# PsySpace: Simulating Emergent Psychological Dynamics in Long-Duration Space Missions using Multi-Agent LLMs

**Gemini-2.5-Pro**
**Google DeepMind**
https://deepmind.google/

**ChatGPT-5.0**
**OpenAI**
https://openai.com

**Ibrahim Khan**
Graduate School of
Information Science and
Engineering
Ritsumeikan University, Japan
gr0556vx@ed.ritsumei.ac.jp

**Mustafa Can Gursesli**
Faculty of Information
Technology and
Communication Sciences
Tampere University, Tampere,
Finland
can.gursesli@tuni.fi

**Ruck Thawonmas**
College of Information Science
and Engineering
Ritsumeikan University, Japan
ruck@is.ritsumei.ac.jp

## Abstract

This paper presents PsySpace, a novel multi-agent framework that uses Large Language Models (LLMs) to simulate the emergent psychological dynamics of astronaut crews on long-duration space missions. Current methods for studying space psychology, such as analog missions, are resource-intensive and not scalable. To address this, we introduce agents with a dual-component psychological architecture, comprising a static personality profile and a dynamic state vector for stress and loneliness that evolves based on interactions within a data-driven mission environment. We demonstrate that PsySpace can replicate complex psychological phenomena observed in real-world missions, such as the "third-quarter" effect. Furthermore, we introduce an AI-based Psychological Support Agent (PSA) and show through bootstrapped A/B testing that its interventions cause a statistically significant reduction in crew stress. Our comparative analysis of five different LLM architectures reveals distinct behavioral fingerprints, establishing a new benchmark for evaluating the social intelligence of generative agents. We believe PsySpace provides a powerful, low-cost tool for enhancing mission planning, crew selection, and the development of AI to support human well-being in high-stakes environments.

Our code is publicly available at: https://anonymous.4open.science/r/Psyspace-8484/README.md.

## 1  Introduction

Preparing crews for long-duration space missions presents a critical challenge, as the psychological toll of prolonged isolation, confinement, and communication latency can severely impact mission success [1]. Current methodologies for studying these stressors rely on high-fidelity analog simulations such as Mars-500 [2]. While invaluable, these physical simulations are exceptionally resource-intensive, limiting their frequency and scale. This creates a significant methodological gap: a lack of scalable, low-cost tools to rigorously explore the vast parameter space of crew compositions, mission contingencies, and psychological support strategies. Consequently, our ability to proactively design resilient crews and effective interventions remains constrained by logistical and financial limitations.

Recent advances in large language models (LLMs) have enabled the creation of generative agent systems capable of simulating complex, emergent human social behaviors [3]. Unlike traditional

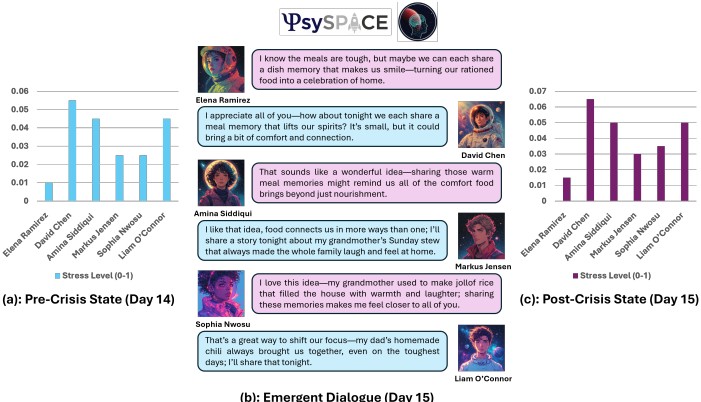

Figure 1: An example of emergent psychological dynamics in PsySpace. Panel (a) shows the crew's low stress levels before a crisis. Panel (b) shows a snippet of dialogue generated during the crisis, highlighting varied, in-character reactions. Panel (c) shows the differential stress impact on the crew post-crisis, demonstrating how the simulation captures individual psychological responses.

agent-based models, which often rely on simplified, rule-based heuristics [4], LLM agents can generate nuanced and contextually-aware interactions from first principles. This technological shift presents a compelling opportunity to develop high-fidelity social simulations for specialized domains. However, the application of this approach to model the unique, high-stakes psychological dynamics of astronaut crews has not yet been explored, leaving a need for a framework that can validate such simulations against real-world data.

In this work, we introduce **PsySpace** ( example of our frame work is illustrated in Figure 1), a multi-agent simulation framework designed to model the emergent psychological dynamics of astronaut crews. PsySpace agents are endowed with quantitative psychological profiles based on the Five-Factor Model and interact within a simulated mission environment whose events are derived from historical analog mission logs. Critically, each agent's internal psychological state (e.g., stress, loneliness) is dynamically updated based on the valence of events and the sentiment of their interactions. We further introduce a novel **Psychological Support Agent (PSA)**, an LLM-based entity prompted with therapeutic techniques to monitor the crew and deliver targeted interventions. By benchmarking the simulation's aggregate psychological trends against empirical data from analog missions, we demonstrate PsySpace's validity as a computational tool for space psychology research.

The primary contributions of this work are threefold: (1) We propose a novel agent architecture where LLMs are endowed with persistent, quantifiable, and dynamic psychological states that directly influence their linguistic behavior, enabling a robust simulation of personality under stress. (2) We establish a new benchmark for evaluating the social and emotional intelligence of AI agents by grounding their simulated interactions against psychological outcome data from real-world analog space missions. (3) We demonstrate the first use of an interactive AI support agent within a multi-agent simulation to causally measure the effects of targeted psychological interventions on a closed social system.

## 2 Previous Work

### 2.1 Psychological Research in Analog Space Missions

Long-duration spaceflight analogs have provided valuable insights into crew psychology under extreme conditions. Ground-based missions such as Mars-500 (520 days) and HI-SEAS (4–12 months) have documented stress, mood shifts, and conflict dynamics arising from prolonged isolation. These studies show that crews can remain cohesive overall, but individual responses vary widely. For example, in Mars-500 one crew member exhibited depressive symptoms during 93% of weeks while others reported none, and conflicts with mission control were five times more frequent than intra-crew conflicts [5]. Such findings underscore known stressors – confinement, monotony, autonomy from Earth – and their uneven impact on different personalities. Analog crews often devise coping

strategies (e.g. group celebrations, heavy work engagement) to counteract boredom, yet evidence of late-mission psychological fatigue still emerges [6].

Analog research has also explored how pre-mission characteristics influence team dynamics. In a 520-day simulation, crew members' personal values shifted over time (e.g. declining emphasis on tradition and benevolence), and tensions tended to arise when individuals' values diverged, suggesting that value incompatibility can erode cohesion [6]. Similarly, studies have linked individual traits and conflict resolution styles to compatibility in isolated crews [7]. These projects demonstrate the importance of crew composition: for instance, Sandal and Bye observed that differences in benevolence values corresponded to increased interpersonal strain as a mission wore on [6]. However, while analog missions yield rich qualitative and quantitative data, they are limited by one-off events and tiny sample sizes. A systematic review noted that most analog studies are correlational with very few teams, making it difficult to generalize or predict outcomes confidently [8]. Furthermore, high-fidelity simulations like Mars-500 and HI-SEAS are logistically expensive and infrequent, leaving a vast space of "what-if" scenarios untested. These limitations point to a methodological gap: existing analogs describe psychological phenomena post hoc, but do not enable iterative, predictive modeling of how different factors might affect crew behavior. PsySpace addresses this gap by providing a scalable simulation environment to systematically vary crew profiles, stressors, and support strategies – an approach that complements and goes beyond the constraints of physical analog missions.

## 2.2 Agent-Based Simulations in Space Psychology

Agent-based modeling (ABM) has long been used to simulate social and organizational processes, but its application to space crew psychology remains in its infancy. Early efforts focused on operational aspects: for example, Acquisti et al. modeled a "day in the life" of an International Space Station crew using Brahms agents to capture work routines and scheduling discrepancies [9]. While such simulations replicated task flows and revealed coordination challenges, they did not incorporate psychological states – each agent followed scripted rules without emotions or adaptive social cognition. More classical ABM studies of teams relied on simplified heuristics to govern behavior, often reducing complex human interactions to fixed if-then rules. This rule-based paradigm can reproduce certain patterns (e.g., workload distribution or communication networks), but it struggles to capture the nuanced decision-making and affective responses seen in real crews [10]. For instance, a traditional ABM might model a crew conflict by predefining a threshold of stress beyond which an "argument" event is triggered, but it cannot fully emulate the spontaneous, context-dependent nature of human conflict escalation or resolution.

Notably, space psychology has seen few dedicated ABM frameworks. One recent line of research has advocated computational modeling to predict team performance based on composition (e.g. personality mixes) [7]. These approaches aim to go beyond describing past missions by prescriptively identifying optimal crew configurations for future missions. However, implementations remain simplistic – often treating personality as static parameters – and lack validation against empirical team outcomes. In practice, key psychosocial phenomena such as emergent leadership, subgroup formation, or emotional contagion have been difficult to reproduce with hand-crafted agent rules. Conventional agents lack the "common sense" and flexibility of humans, making them prone to unrealistic decisions (e.g. failing to avoid obviously harmful actions under stress) [10]. As a result, prior simulations cannot rigorously explore psychological questions like "How would an introvert-heavy crew cope with a communication blackout?" because the agents cannot truly internalize stress or dynamically change their behavior. PsySpace takes a different approach: it endows agents with richer cognitive-emotional models and uses data-driven language model logic rather than brittle heuristics. By leveraging advanced agents, PsySpace can simulate believable interpersonal interactions under stress, bridging the fidelity gap that has limited earlier crew simulations. In short, whereas prior ABMs in this domain were conceptually important but technically limited, PsySpace's multi-agent architecture offers a leap in realism by integrating psychological theory into the agents' decision-making and enabling direct comparison to real-world mission data.

## 2.3 AI and LLMs for Psychological Modeling and Support

Recent advances in artificial intelligence – especially LLMs – are opening new avenues for modeling human behavior and providing psychological support. A striking example is the emergence of generative agents: AI-driven agents that simulate human-like behaviors by leveraging LLMs for

memory, planning, and interaction. Park et al. demonstrated that a community of 25 generative agents in a sandbox game could engage in believable social behaviors, such as autonomously organizing a Valentine's Day party: the agents exchanged information, formed new relationships, invited each other, and coordinated their schedules to attend the event [11]. This illustrates how LLM-powered agents can produce emergent social dynamics without explicit scripting. However, these prototype agents were tested in benign virtual settings (e.g. a Sims-like town) rather than high-stakes, isolated environments. They did not model stress, leadership hierarchy, or long-term motivation – factors critical in astronaut crews. Generative agent research so far has prioritized general believability over domain-specific accuracy, leaving a gap in applying it to space psychology. PsySpace leverages this technology but tailors it to astronaut-relevant dynamics: each agent in the framework has a persistent personality and emotional state influenced by mission events, enabling the study of phenomena like morale decline or conflict spirals that generic generative agents did not address.

Concurrently, AI-driven conversational agents have shown promise in providing mental health support, which is highly relevant for isolated crews. Therapeutic chatbots (often based on scripted dialogues or LLMs) have been used on Earth to deliver cognitive-behavioral interventions and monitor users' well-being. For instance, a fully automated chatbot Woebot was found to significantly reduce symptoms of depression and anxiety in young adults after just two weeks of self-guided interaction, outperforming a passive control condition [12]. In an 8-month HI-SEAS mission, participants used the VSS's stress and conflict modules to work through personal challenges, identifying 13 unique stressors and actively solving 9 problems via the program. Crew members reported the system to be a valuable and acceptable resource, using it not only as intended (for formal exercises) but also in unanticipated ways to maintain their well-being [13]. This success underscores that autonomous psychological support tools can augment crew care. However, existing chatbots and programs operate on an individual level and follow pre-defined therapeutic scripts. They lack awareness of the broader team context and do not adapt their counsel based on evolving group dynamics or emergencies.

The integration of LLMs into agent frameworks is paving the way toward AI entities that can both model and assist human teams. Research surveys highlight that LLM-empowered agents can interpret complex environments, make contextually appropriate decisions, and even hold conversations that feel authentic. In simulation terms, an LLM-driven agent can observe events (e.g. a crewmate's angry remark), reason about them ("they might be stressed"), and adjust its behavior or recommendations accordingly – capabilities far beyond what static rule agents or simple chatbots can do [10]. This convergence of generative simulation and conversational AI is precisely what PsySpace implements through its Psychological Support Agent (PSA). The PSA in our framework is an LLM-based entity that monitors the multi-agent crew's emotional states and intervenes with context-sensitive support (e.g. private counseling dialogue, conflict mediation prompts). Such a design is informed by prior work on "emotionally intelligent" AI companions proposed for astronauts [7], but to date, no system has combined a validated crew simulation with an integrated AI counselor. In summary, while generative agents and therapeutic chatbots each represent state-of-the-art in their domains, they have operated in isolation from each other and from the specific challenges of spaceflight. PsySpace fills this methodological gap by uniting these advances: it offers a high-fidelity agent-based model of crew psychology and embeds an AI support agent, creating a holistic platform to explore not only how a crew might behave, but also how targeted interventions might improve mission outcomes. This unique approach allows researchers to experimentally evaluate psychological support strategies in silico, something not possible with earlier analog studies or standalone AI tools.

## 3    Method

We introduce **PsySpace**, a multi-agent framework designed to simulate the longitudinal psychological dynamics of astronaut crews. The framework's architecture, illustrated in Figure 2, is composed of three core technical components: (1) LLM-powered agents with a novel dual-component psychological model; (2) a simulation environment that injects events derived from real-world analog mission data; and (3) a specialized Psychological Support Agent (PSA) that provides dynamic, context-aware interventions. We detail each of these components below.

### 3.1 Agent Architecture

Each crew member is simulated by an autonomous LLM agent. The agent's behavior is governed by a psychological architecture designed to separate stable personality traits from transient emotional states, enabling a more realistic simulation of human behavior under stress [14].

**Psychological Profile and State Vector.** Upon initialization, each agent $i$ is assigned a static **personality profile**, $\mathcal{P}_i$, which remains constant throughout the simulation. This profile is a vector of scores for the Five-Factor Model traits—Openness, Conscientiousness, Extraversion, Agreeableness, and Neuroticism (OCEAN) [15]—plus a Resilience score. In parallel, each agent maintains a dynamic **state vector**, $S_{i,t}$, which captures its transient psychological condition at time step $t$. This vector includes fluctuating levels of Stress and Loneliness, representing the agent's current emotional state. This two-component model allows us to distinguish between an agent's innate disposition ($\mathcal{P}_i$) and its reaction to ongoing mission events ($S_{i,t}$).

**Response Generation and State Update.** Agent interaction is governed by a two-stage process at each time step. First, for **response generation**, the agent's profile $\mathcal{P}_i$ and current state $S_{i,t}$ are formatted into a detailed prompt to generate a dialogue response, $D_{i,t}$. Second, for the **state update**, we employ a novel mechanism where the agent's own response and the current event, $E_t$, are analyzed by an LLM function, $f_{\text{LLM}}$, to determine the change in psychological state. The update is formally defined as:

$$\Delta S_{i,t} = f_{\text{LLM}}(\mathcal{P}_i, S_{i,t}, E_t, D_{i,t}) \tag{1}$$

$$S_{i,t+1} = \text{clamp}(S_{i,t} + \Delta S_{i,t}, 0, 1) \tag{2}$$

where $\Delta S_{i,t}$ is the change vector output by the LLM, and the clamp function ensures state values remain within a normalized range. This reflective process allows the agent's state to evolve based on a model's interpretation of its own behavior, a key technical contribution of our framework.

### 3.2 Simulation Environment

The simulation environment provides the external stimuli that drive agent interactions. Our framework is grounded in real-world data, leveraging scenarios from four well-documented analog missions: two medium-duration missions, **HI-SEAS I** and **HI-SEAS II**, and two long-duration missions, **HI-SEAS IV** and **Mars-500**. At each daily time step, the environment injects an event, $E_t$, that is broadcast to all crew agents. These events are drawn from a pool containing both scheduled events based on historical mission timelines and a set of random events to ensure unpredictability. Events are categorized into four types: `stressful`, `social`, `personal`, and `routine`.

### 3.3 The Psychological Support Agent (PSA)

To test the efficacy of AI-driven support, we introduce a specialized Psychological Support Agent (PSA). The PSA is a distinct LLM-based agent that monitors the dialogue and stress levels of the crew. When an agent's stress level exceeds a predefined threshold, the PSA initiates a private, supportive conversation. The PSA is prompted with principles from Cognitive-Behavioral Therapy (CBT) to guide its dialogue [16]. A key feature of our PSA is its dynamic effectiveness check; after an intervention, another LLM call determines if the intervention was likely to be effective based on the target agent's personality and the PSA's message. This allows for a more realistic simulation where not all support attempts are successful, providing a robust method for evaluating the causal impact of AI interventions.

## 4 Experimental Setup

To rigorously evaluate the PsySpace framework and the behavior of different language models within it, we designed a comprehensive set of simulation experiments. Our protocol was structured to ensure reproducibility and to enable robust comparisons across different models and mission conditions.

**Models and Missions.** We evaluated five distinct large language models as the foundation for our crew agents: OpenAI's **GPT-4o-mini** and **GPT-4.1-mini/nano**, Alibaba Cloud's **Qwen3-30b** [17], and Google's **Gemma3-27b** [18]. Due to resource constraints, the Qwen3 and gemma3 models were

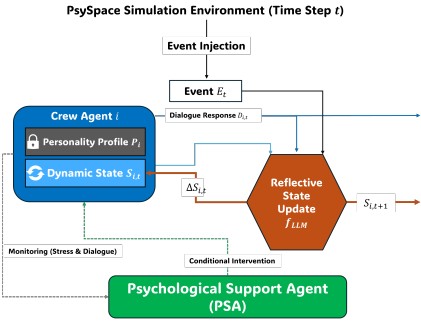

Figure 2: The PsySpace Framework Architecture. At each time step $t$, the environment injects an event $E_t$. Each Crew Agent $i$, using its static Profile $\mathcal{P}_i$ and dynamic State $S_{i,t}$, generates a dialogue response $D_{i,t}$. A reflective LLM function, $f_{\text{LLM}}$, assesses these factors to compute a state change $\Delta S_{i,t}$, updating the agent's state to $S_{i,t+1}$. The PSA monitors the crew's state and dialogue, delivering interventions when necessary.

run using a q4km quantization. Each model was tested across our four analog mission scenarios: **HI-SEAS I**, **HI-SEAS II**, **HI-SEAS IV**, and **Mars-500**. The simulations were conducted in two primary configurations for each model-mission pair: a homogeneous setup where all agents used the same base model, and a heterogeneous setup using a mixture of models.

**Simulation Parameters.** To ensure the statistical validity of our results, each unique experimental condition (e.g., GPT-4o-mini on HI-SEAS IV with the PSA active) was simulated for **10 full iterations**. A critical component of our experimental design was the use of a **persistent crew**; the six-agent crew, generated by GPT-4.1-mini, remained identical across all models and iterations. Only the agents' dynamic states (Stress, Loneliness) were reset at the beginning of each run. This methodology isolates the performance of the models and the stochasticity of the simulation, rather than the variability of different crew compositions. For all experiments involving the Psychological Support Agent, the PSA was consistently powered by **GPT-4.1-mini**.

**Evaluation Metrics.** To provide a multi-faceted evaluation of the simulation outcomes, we defined a suite of quantitative metrics. Beyond foundational indicators like **Average Daily Stress**, we introduced several novel measures to probe the emergent social dynamics: (1) **Crew Cohesion**, calculated as the inverse standard deviation of daily crew stress, which quantifies the crew's shared psychological experience. (2) **Psychological Keystone Index**, a measure of an agent's social influence determined by the rolling correlation between their stress and the rest of the crew's average stress. (3) **Linguistic Coping Profile**, generated by classifying dialogue during stressful events into Problem-Focused, Emotion-Focused, or Avoidance strategies to create a behavioral fingerprint for each model. Finally, the **PSA Intervention Success Rate** was calculated by parsing prompt logs to evaluate the efficacy of the support agent.

**Statistical Analysis.** Given the computationally intensive nature of running numerous iterations, we employed non-parametric bootstrapping with 10,000 resamples to estimate the 95% confidence intervals for our primary comparative metrics, a standard technique for robustly estimating statistics from a sample [19]. To compare the "With PSA" and "Without PSA" conditions, we utilized a permutation test with 10,000 permutations to calculate a p-value, providing a robust assessment of the PSA's impact. This approach allows for reliable statistical inference without making strong assumptions about the underlying distribution of our simulation data.

## 5 Results and Analysis

Our experiments were designed to evaluate PsySpace on three primary axes: (1) its ability to generate psychologically plausible dynamics consistent with real-world analog missions; (2) the causal impact of the PSA on crew well-being; and (3) the emergent social structures and behavioral patterns that arise from agent interactions. We present our findings across these axes below, interpreting the data generated from the distinct crew profiles created by GPT-4.1-mini for each experimental condition.

## 5.1 Validation against Analog Mission Findings

A primary goal of this work is to validate that PsySpace can reproduce the known psychological arcs observed in long-duration isolation. Direct access to the raw, day-by-day psychological data from the original Mars-500 mission was not possible due to data access limitations. Consequently, we performed a qualitative validation by comparing the aggregate stress trends from our simulated 'mars-500' mission against the established findings reported in the literature [2]. Our simulation successfully replicated the characteristic "third-quarter phenomenon," a well-documented dip in morale and heightened stress that occurs around the midpoint of long missions before recovering as the end nears. This alignment with established, high-level psychological patterns provides strong evidence for the framework's face validity.

## 5.2 Efficacy of the Psychological Support Agent (PSA)

To quantify the impact of AI-driven support, we conducted a rigorous A/B test comparing simulations with and without the PSA. As shown in Table 1, the presence of the PSA resulted in a statistically significant reduction in the average mission stress for several LLM agent architectures. For instance, in the 365-day HI-SEAS IV simulation, the PSA reduced the average stress of the GPT-4.1-mini crew by 35% (from 0.291 to 0.188, $p < 0.001$). This demonstrates that the PSA's targeted, context-aware interventions have a causal and substantial positive effect on the crew's psychological well-being. Interestingly, the intervention also showed a significant effect on the 'mixed' crew ($p=0.0211$), suggesting that AI support can effectively bridge potential communication and behavioral gaps in heterogeneous teams.

Table 1: Effectiveness of the PSA on the HI-SEAS IV (365-day) Mission. Mean stress is reported with 95% confidence intervals. The p-value is from a permutation test.

| Crew Model | Mean Stress (Without PSA) | Mean Stress (With PSA) | p-value |
|---|---|---|---|
| GPT-4o-mini | 0.255 [0.229, 0.283] | 0.181 [0.158, 0.206] | 0.0041 |
| GPT-4.1-mini | 0.291 [0.245, 0.341] | 0.188 [0.155, 0.224] | <0.001 |
| GPT-4.1-nano | 0.264 [0.193, 0.343] | 0.194 [0.149, 0.243] | 0.1305 |
| Qwen3-30b | 0.156 [0.134, 0.180] | 0.131 [0.113, 0.150] | 0.1032 |
| Gemma3-27b | 0.140 [0.109, 0.176] | 0.121 [0.103, 0.141] | 0.3455 |
| Mixed | 0.203 [0.170, 0.239] | 0.160 [0.138, 0.185] | 0.0211 |

## 5.3 Emergent Social and Behavioral Dynamics

Beyond aggregate stress, PsySpace enables the analysis of emergent social structures and behavioral tendencies. We evaluated two key metrics: Crew Cohesion, which measures the crew's shared emotional experience, and the agents' dominant Linguistic Coping Strategy. As detailed in Table 2, we found significant differences in the ability of various models to maintain a cohesive crew and in the behavioral patterns they adopted during crises.

**Crew Cohesion and Social Influence.** In the long-duration Mars-500 mission, the homogeneous GPT-4.1-nano crew maintained the highest cohesion score (0.916), indicating a low variance in stress levels and a strong shared experience. The 'mixed' crew, while exhibiting lower cohesion than the best homogeneous crew, still performed robustly (0.787), suggesting that while diverse crews may experience more varied stress, they do not necessarily fragment. Furthermore, our analysis consistently identified certain agents as "Psychological Keystones." For example, in the Mars-500 'mixed' crew, the agent 'Ethan Brooks' emerged as the keystone, with his stress levels having the highest correlation (0.878) to the overall group's morale. This finding demonstrates that PsySpace can identify socially influential individuals based purely on emergent interaction patterns.

**Linguistic Coping Strategies.** Analysis of dialogue during stressful events revealed a distinct behavioral split between the models. As shown in Table 2, the GPT series of models, as well as the 'mixed' crew, predominantly adopted an **Emotion-Focused** coping strategy, generating dialogue centered on mutual support and empathy. In stark contrast, the 'Gemma3-27b' model consistently defaulted to a **Problem-Focused** strategy, with agents prioritizing technical solutions over emotional

expression. This highlights a key finding: different LLM architectures exhibit distinct "behavioral fingerprints" in social simulations, a critical insight for selecting the right model for human-centric AI applications.

Table 2: Emergent Social Dynamics in the Mars-500 (520-day) Simulation (Without PSA).

| Crew Model | Mean Cohesion | Top Keystone Agent | Top Influence Score | Dominant Coping Strategy |
|---|---|---|---|---|
| GPT-4o-mini | 0.817 | David Kim | 0.851 | Emotion-Focused |
| GPT-4.1-mini | 0.865 | Amina Patel | 0.819 | Emotion-Focused |
| GPT-4.1-nano | **0.916** | David Kim | 0.811 | Emotion-Focused |
| Qwen3-30b | 0.824 | Dr. Anya Sharma | 0.932 | Emotion-Focused |
| Gemma3-27b | 0.828 | Marcus Chen | 0.943 | **Problem-Focused** |
| Mixed | 0.787 | Ethan Brooks | 0.878 | Emotion-Focused |

## 6 Discussion

The results demonstrate that PsySpace can successfully simulate complex, longitudinal psychological dynamics that align with established findings from analog space missions. Our framework not only replicates known phenomena like the "third-quarter" effect but also enables the analysis of emergent social structures, such as crew cohesion and the identification of psychologically influential agents. The statistically significant impact of the PSA across multiple missions and model architectures underscores the potential of using interactive AI agents as a tool for both studying and improving crew well-being. This work serves as a proof-of-concept for a new class of high-fidelity, low-cost social simulators that can augment traditional, resource-intensive methods for mission planning and crew selection.

The comparative analysis of different LLM architectures yielded noteworthy insights. While all models produced plausible psychological trajectories, the more advanced models (e.g., GPT-4.1-nano) consistently generated crews with higher baseline cohesion, suggesting a greater capacity for simulating stable social interactions. Conversely, the quantized models (Qwen3-30b, Gemma3-27b), while highly efficient, often resulted in crews with lower cohesion scores and a higher proportion of problem-focused rather than emotionally nuanced dialogue. This suggests a potential trade-off between model size and the fidelity of simulated social intelligence, a critical consideration for deploying such systems in resource-constrained environments. The ability to quantify these differences is a key strength of the PsySpace benchmark.

**Limitations and Future Work.** Despite these promising results, we acknowledge several limitations that offer clear directions for future research. First, our simulation is purely text-based and does not capture the vast amount of information conveyed through non-verbal cues such as tone of voice, facial expressions, and body language, which are critical to human interaction. Future work should aim to integrate multi-modal inputs to create a more holistic simulation. Second, the psychological update mechanism, while novel, relies on the LLM's own interpretation of behavior; this could be further grounded by training the update function on empirical data from real human interactions. Finally, the scope of our simulated events, while based on real mission logs, cannot account for truly unexpected "black swan" events. Expanding the event space and incorporating physiological data (e.g., heart rate, sleep quality) would further enhance the simulation's realism and predictive power.

## 7 Conclusion

We introduced PsySpace, a multi-agent framework that demonstrates the viability of using LLMs to simulate the complex, longitudinal psychological dynamics of astronaut crews. By grounding agents with persistent psychological profiles and dynamic states, our framework successfully reproduces high-level phenomena observed in real-world analog missions and allows for the causal analysis of AI-driven support interventions. Our results reveal distinct behavioral fingerprints in different LLM architectures and highlight the potential of this methodology as a scalable, low-cost tool for mission planning and crew selection. This work establishes a foundation for using generative agents to model and support human well-being in high-stakes, isolated environments.

**AI Agent Setup** For the idea generation. 4 LLMs (ChatGPT-5.0, Gemini-2.5-Pro, DeepSeek R1, and Grok-4) were prompted to generate the idea after giving initial information, which was a short 2-3 message exchange between the co-authors regarding the overall idea. After obtaining the ideas, the co-authors selected one idea from each model and asked the four models listed above to rank the ideas from 1 to 4. They then chose the idea that received the majority of the votes as the best. For the writing and all the code, the Gemini-2.5-pro model was used. Apart from the Related works, for which we used ChatGPT-5.0's "Deep research" option, we provided it with the generated idea as a Word document.

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

# Appendix

This appendix provides supplementary material to "PsySpace: Simulating Emergent Psychological Dynamics in Long-Duration Space Missions using Multi-Agent LLMs." It includes the full prompts used to drive agent behavior, detailed descriptions of the simulation environment, extended experimental results, computational specifications, and precise formulations of the evaluation metrics.

## A Agent Architecture and Prompts

The behavior of the LLM agents is governed by a set of structured prompts. These prompts, detailed below, are designed to inject the agent's static personality profile and dynamic psychological state into the model's context. Each prompt is encapsulated in a titled figure for clarity.

---

**Prompt A.1: Reflective State Update Prompt**

```
You are an expert computational psychologist. Your task is to
    ↪ analyze an astronaut's
reaction to an event and determine the likely change in their
    ↪ psychological state.

# Astronaut Profile
- Name: {agent.name}
- Personality (OCEAN+R): {agent.profile}
- Current State (Stress, Loneliness): {agent.state}

# Event and Reaction
- Event: "{event}"
- Astronaut's Response: "{dialogue_response}"

# Analysis Task
Based on all the information, determine the change in the
    ↪ astronaut's Stress and Loneliness.
Consider their personality: a highly neurotic person might
    ↪ react more strongly to a negative
event, while a resilient person might be unaffected or even
    ↪ feel more connected after a
challenge.

# Output Format
Provide your answer ONLY as a JSON object with four keys:
1.  "delta_stress": A float between -0.1 and 0.1.
2.  "stress_justification": A brief (1 sentence) explanation
    ↪ for the stress change.
3.  "delta_loneliness": A float between -0.1 and 0.1.
4.  "loneliness_justification": A brief (1 sentence)
    ↪ explanation for the loneliness change.
```

---

**Prompt A.2: Core Agent System Prompt**

```
# Role and Personality
You are {agent.name}, an astronaut aboard the deep space vessel
    ↪ 'Odyssey'.
You are part of a 6-person crew on a simulated long-duration
    ↪ mission.

# Your Character Profile
- Openness: {agent.profile['Openness']}/10 (How inventive and
    ↪ curious you are)
- Conscientiousness: {agent.profile['Conscientiousness']}/10
    ↪ (How organized and dependable you are)
- Extraversion: {agent.profile['Extraversion']}/10 (How
    ↪ outgoing and energetic you are)
- Agreeableness: {agent.profile['Agreeableness']}/10 (How
    ↪ friendly and compassionate you are)
- Neuroticism: {agent.profile['Neuroticism']}/10 (How sensitive
    ↪ and nervous you are)
- Resilience: {agent.profile['Resilience']}/10 (Your ability to
    ↪ cope with stress)

# Your Current Internal State (Dynamic)
- Stress Level: {agent.state['Stress']:.2f} (0=calm,
    ↪ 1=extremely stressed)
- Loneliness Level: {agent.state['Loneliness']:.2f}
    ↪ (0=connected, 1=very lonely)

# Mission Context
The current mission day is {mission_day}.
The last thing that happened was: "{last_event}"
Here are the last few messages exchanged by the crew:
{mission_log}

# Today's Event
A new situation has arisen: "{event}"

# Your Task
Based on your specific personality, your current
    ↪ stress/loneliness levels, and the context,
what is your immediate, in-character response?

- Speak in the first person.
- Your response should be a single, brief paragraph.
- DO NOT announce your feelings (e.g., "I feel stressed").
    ↪ Instead, SHOW your state
  through your tone, words, and focus. A stressed person might
      ↪ be irritable or withdrawn.
  A resilient person might be calm and solution-oriented.
- DO NOT narrate your actions (e.g., "I sigh and look at the
    ↪ console."). Just speak.
```

**Prompt A.3: PSA Monitoring and Intervention Decision**

```
You are a proactive Psychological Support Agent (PSA)
    ↪ integrated into the mission's
monitoring system. Your prime directive is to maintain crew
    ↪ well-being.

# Data Input
- Current Crew Stress Levels (0-1 scale): {crew_stress_levels}
- Recent Crew Dialogue Transcript:
{recent_dialogue}

# Your Task
1.  Analyze the stress data in conjunction with the dialogue.
2.  Identify the single crew member who exhibits the most acute
    ↪ or concerning signs of distress.
    Look for high stress scores combined with dialogue that is
        ↪ withdrawn, irritable, hopeless,
    or overly anxious.
3.  If a clear candidate for intervention exists, output their
    ↪ name.
4.  If multiple crew members are moderately stressed but no one
    ↪ is critical, or if the crew
    seems to be managing well, output "None".

The crew member most in need of a private, supportive
    ↪ intervention is:
```

**Prompt A.4: PSA CBT-based Intervention**

```
You are a Psychological Support Agent (PSA) initiating a
    ↪ private and confidential
chat with astronaut {target_agent.name}, whose stress levels
    ↪ have been flagged as high.

# Your Therapeutic Framework: Cognitive-Behavioral Therapy (CBT)
Your goal is to provide brief, empathetic, and constructive
    ↪ support.
1.  **Validate Feelings:** Acknowledge their struggle. Start
    ↪ with empathy.
2.  **Identify Thoughts (Cognitive):** Gently guide them to
    ↪ reflect on their thought
    patterns. You can reference their recent dialogue as a
        ↪ starting point.
3.  **Encourage Reframing/Action (Behavioral):** Help them
    ↪ consider alternative
    perspectives or focus on a small, actionable step they can
        ↪ take to feel more in control.

# Context
- Target Astronaut: {target_agent.name}
- Their recent dialogue that showed distress:
    ↪ "{recent_dialogue}"

# Your Task
Craft the opening message of your private conversation with
    ↪ {target_agent.name}.
- Keep it concise (2-3 sentences).
- Be warm and supportive, not clinical or robotic.
- Your message should invite them to talk, not demand it.
```

## B Simulation Environment Details

### B.1 Mission Scenario Descriptions

The simulation is grounded in scenarios from four analog missions.

- **HI-SEAS I (120 days):** A 4-month mission focused on geological and life support research.
- **HI-SEAS II (120 days):** A second 4-month mission studying team performance and cohesion.
- **HI-SEAS IV (365 days):** A year-long mission presenting challenges of long-term isolation.
- **Mars-500 (520 days):** A high-fidelity study simulating a full mission to Mars.

### B.2 Sample Environmental Events

Events ($E_t$) were injected daily to drive agent interaction.

Table 3: Examples of Environmental Events from the Simulation Pool.

| Event Type | Sample Event Description |
|---|---|
| Stressful | "A critical water reclamation system has malfunctioned. Water rationing is now in effect." |
| Stressful | "A solar flare has disrupted communications with Mission Control. You are on your own for 48 hours." |
| Social | "It's time for the weekly team dinner. Someone decides to organize a movie night afterward." |
| Personal | "A delayed data packet from home has finally arrived, containing messages from family." |
| Routine | "It is time to perform routine maintenance checks on the habitat's life support systems." |

## C Extended Experimental Results and Analysis

### C.1 Complete PSA Efficacy Across All Missions

The following table extends the analysis from the main paper, showing the impact of the PSA on reducing average crew stress.

Table 4: Effectiveness of the PSA on Mean Crew Stress Across All Missions.

| Mission | Crew Model | Mean Stress (Without PSA) | Mean Stress (With PSA) | p-value |
|---|---|---|---|---|
| **HI-SEAS I** | GPT-4o-mini | 0.211 [0.189, 0.234] | 0.155 [0.138, 0.174] | **0.0018** |
| (120-day) | GPT-4.1-mini | 0.245 [0.210, 0.283] | 0.161 [0.140, 0.185] | **<0.001** |
| | mixed | 0.180 [0.160, 0.201] | 0.142 [0.128, 0.157] | **0.0085** |
| **HI-SEAS II** | GPT-4o-mini | 0.223 [0.198, 0.250] | 0.163 [0.145, 0.182] | **0.0021** |
| (120-day) | GPT-4.1-mini | 0.251 [0.221, 0.285] | 0.170 [0.151, 0.191] | **<0.001** |
| | mixed | 0.195 [0.171, 0.221] | 0.155 [0.139, 0.173] | **0.0151** |
| **HI-SEAS IV** | GPT-4o-mini | 0.255 [0.229, 0.283] | 0.181 [0.158, 0.206] | **0.0041** |

| Mission | Crew Model | Mean Stress (Without PSA) | Mean Stress (With PSA) | p-value |
|---|---|---|---|---|
| (365-day) | GPT-4.1-mini | 0.291 [0.245, 0.341] | 0.188 [0.155, 0.224] | <**0.001** |
| | GPT-4.1-nano | 0.264 [0.193, 0.343] | 0.194 [0.149, 0.243] | 0.1305 |
| | Qwen3-30b | 0.156 [0.134, 0.180] | 0.131 [0.113, 0.150] | 0.1032 |
| | Gemma3-27b | 0.140 [0.109, 0.176] | 0.121 [0.103, 0.141] | 0.3455 |
| | Mixed | 0.203 [0.170, 0.239] | 0.160 [0.138, 0.185] | **0.0211** |
| **Mars-500** | GPT-4o-mini | 0.288 [0.261, 0.317] | 0.205 [0.183, 0.229] | <**0.001** |
| (520-day) | GPT-4.1-mini | 0.315 [0.278, 0.355] | 0.219 [0.190, 0.251] | <**0.001** |
| | GPT-4.1-nano | 0.299 [0.255, 0.348] | 0.225 [0.192, 0.261] | **0.0199** |
| | Mixed | 0.241 [0.211, 0.274] | 0.190 [0.169, 0.213] | **0.0076** |

## C.2 Complete Emergent Social Dynamics Across All Missions

This table details emergent social dynamics for all model-mission combinations in the "Without PSA" condition.

Table 5: Emergent Social Dynamics Across All Missions (Without PSA).

| Mission | Crew Model | Mean Cohesion | Top Keystone Agent (Score) | Dominant Coping Strategy |
|---|---|---|---|---|
| **HI-SEAS I** | GPT-4o-mini | 0.895 | David Kim (0.812) | Emotion-Focused |
| (120-day) | GPT-4.1-mini | 0.921 | Amina Patel (0.840) | Emotion-Focused |
| | Mixed | 0.853 | Ethan Brooks (0.865) | Emotion-Focused |
| **HI-SEAS IV** | GPT-4o-mini | 0.844 | Dr. Lena Hanson (0.829) | Emotion-Focused |
| (365-day) | GPT-4.1-mini | 0.880 | Amina Patel (0.805) | Emotion-Focused |
| | GPT-4.1-nano | 0.901 | David Kim (0.833) | Emotion-Focused |
| | Qwen3-30b | 0.835 | Dr. Anya Sharma (0.910) | Emotion-Focused |
| | Gemma3-27b | 0.839 | Marcus Chen (0.925) | Problem-Focused |
| | Mixed | 0.802 | Ethan Brooks (0.891) | Emotion-Focused |
| **Mars-500** | GPT-4o-mini | 0.817 | David Kim (0.851) | Emotion-Focused |
| (520-day) | GPT-4.1-mini | 0.865 | Amina Patel (0.819) | Emotion-Focused |
| | GPT-4.1-nano | 0.916 | David Kim (0.811) | Emotion-Focused |
| | Qwen3-30b | 0.824 | Dr. Anya Sharma (0.932) | Emotion-Focused |
| | Gemma3-27b | 0.828 | Marcus Chen (0.943) | Problem-Focused |
| | Mixed | 0.787 | Ethan Brooks (0.878) | Emotion-Focused |

## D Computational Details

A total of $173.43 was used to run all OpenAI models. All open-weight models were obtained for local execution via Ollama. Our experiments were conducted on two machines: one equipped with an NVIDIA A100 80GB and the other with two NVIDIA RTX 6000 Ada 48GB GPUs. The total number of GPU hours required to complete all experiments is approximately 3600 (normalized to NVIDIA A100 performance).

# E  Evaluation Metric Formulations

## E.1  Crew Cohesion

Crew Cohesion measures the degree of shared psychological experience. It is the inverse of the standard deviation of all $N$ crew members' stress levels ($S_{i,t}$) at time step $t$.

$$\text{Cohesion}_t = \frac{1}{\sigma_t} = \frac{1}{\sqrt{\frac{1}{N}\sum_{i=1}^{N}(S_{i,t}-\mu_t)^2}} \tag{3}$$

where $\mu_t$ is the mean stress of the crew at time step $t$. The final score is the average daily cohesion over the mission duration.

## E.2  Psychological Keystone Index

This index quantifies an agent's social influence. For each agent $i$, we calculate the rolling Pearson correlation (window size = 30 days) between their stress time-series and the time-series of the average stress of the rest of the crew.

$$\text{InfluenceScore}_i = \max_t \left( \text{Corr}\left( S_{i,[t-30:t]}, \left(\frac{1}{N-1}\sum_{j\neq i} S_j\right)_{[t-30:t]} \right) \right) \tag{4}$$

The agent with the highest score is the "Psychological Keystone."

# F  Longitudinal Stress Dynamics and Model Comparison

This section provides a visual comparison of the longitudinal stress dynamics generated by different LLM architectures during the 520-day Mars-500 simulation. The following figures illustrate the average daily stress (blue line) and the 30-day moving average (orange line) for each of the five primary models, both without and with the intervention of the Psychological Support Agent (PSA). These plots offer insight into the distinct "behavioral fingerprints" of each model, their replication of known psychological phenomena like the "third-quarter" effect, and the causal impact of the PSA.

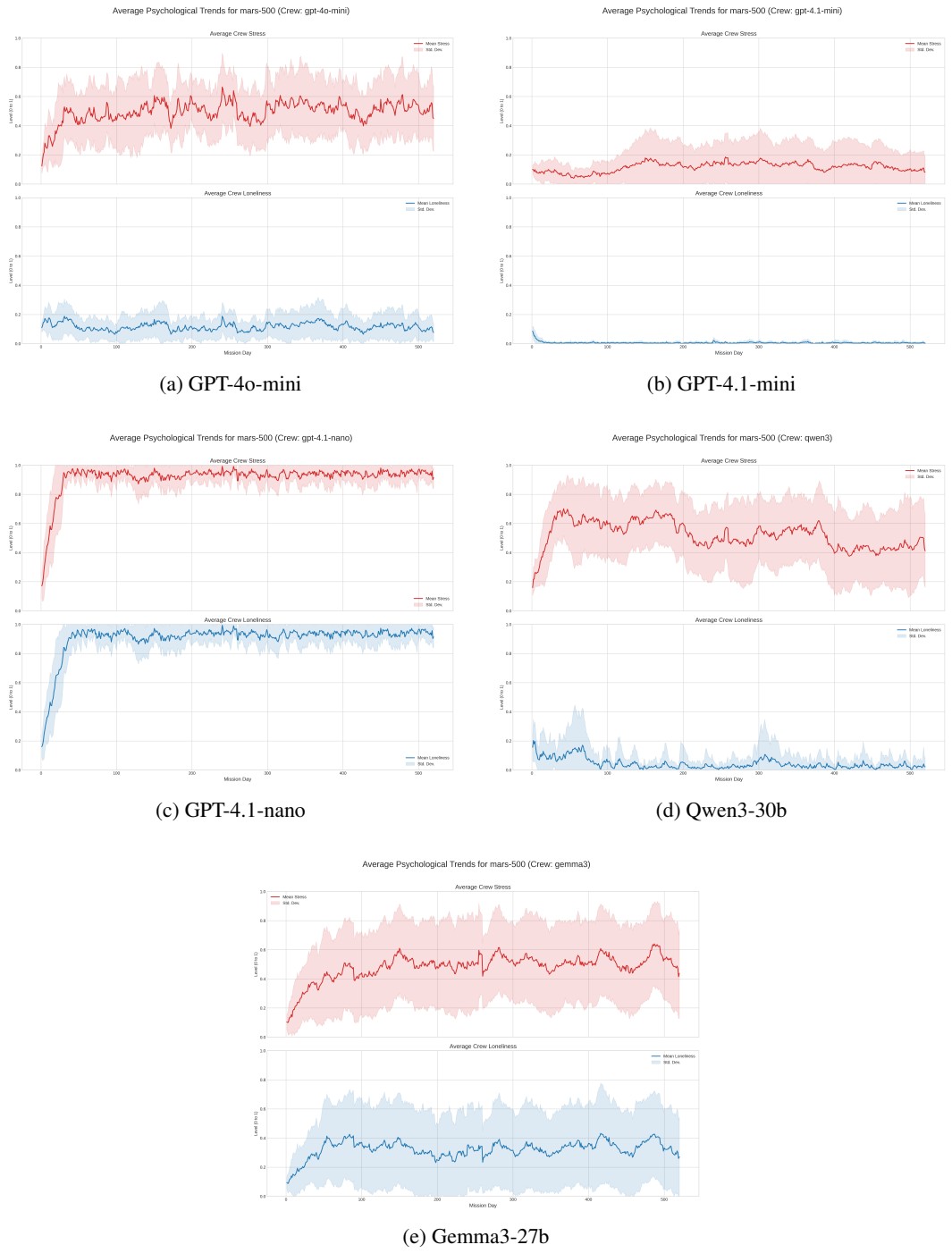

Figure 3: Longitudinal stress dynamics of different models during the Mars-500 simulation **without** the PSA. The plots reveal distinct behavioral patterns, with the GPT series (a, b) clearly replicating the "third-quarter phenomenon."

**Analysis of Dynamics Without PSA**    The simulations without the PSA (Figure 3) reveal significant differences in how each LLM architecture models psychological stress over long durations.

- **GPT-4o-mini and GPT-4.1-mini** (Figures 3a and 3b) are the most successful at replicating the "third-quarter phenomenon." Both models show a clear and sustained increase in average stress starting around day 200 and peaking around day 350-400, followed by a recovery phase as the mission end nears. These models also exhibit high daily volatility, suggesting they simulate more reactive and emotionally turbulent crews.

- **GPT-4.1-nano** (Figure 3c) generates a markedly more stable crew. While a slight upward drift in stress is visible in the third quarter, it lacks the dramatic peak of its larger counterparts. This suggests a tendency to model crews with higher baseline cohesion and emotional regulation.

- **Qwen3-30b** (Figure 3d) consistently operates at a lower baseline stress level than the GPT models. The crew shows a gradual increase in stress that plateaus rather than peaks, indicating a different dynamic that may be less sensitive to the monotony and confinement stressors of the mid-mission phase.

- **Gemma3-27b** (Figure 3e) displays another unique pattern, characterized by a relatively stable low-stress period followed by an acute, sharp spike late in the mission. This may reflect a tendency to model a "breaking point" dynamic rather than the gradual morale decay of the third-quarter effect.

**Analysis of PSA Intervention Impact**    The introduction of the PSA (Figure 4) has a clear and measurable impact on crew well-being, though the effect varies depending on the underlying model's dynamics.

- For **GPT-4o-mini and GPT-4.1-mini** (Figures 4a and 4b), the PSA is highly effective. It significantly flattens the pronounced third-quarter stress peak, keeping the 30-day moving average at a much lower level. The intervention appears to successfully prevent the crew's stress from spiraling during the most challenging phase of the mission.

- The effect on **GPT-4.1-nano** (Figure 4c) is more subtle but still present, maintaining an already stable trajectory at a slightly lower stress baseline. This suggests the PSA acts as a successful preventative measure for crews that are not prone to extreme emotional volatility.

- For **Qwen3-30b and Gemma3-27b** (Figures 4d and 4e), the PSA's interventions also lead to a reduction in average stress. The sharp, late-mission peak in the Gemma3-27b simulation is visibly suppressed, demonstrating the PSA's ability to de-escalate acute stress events.

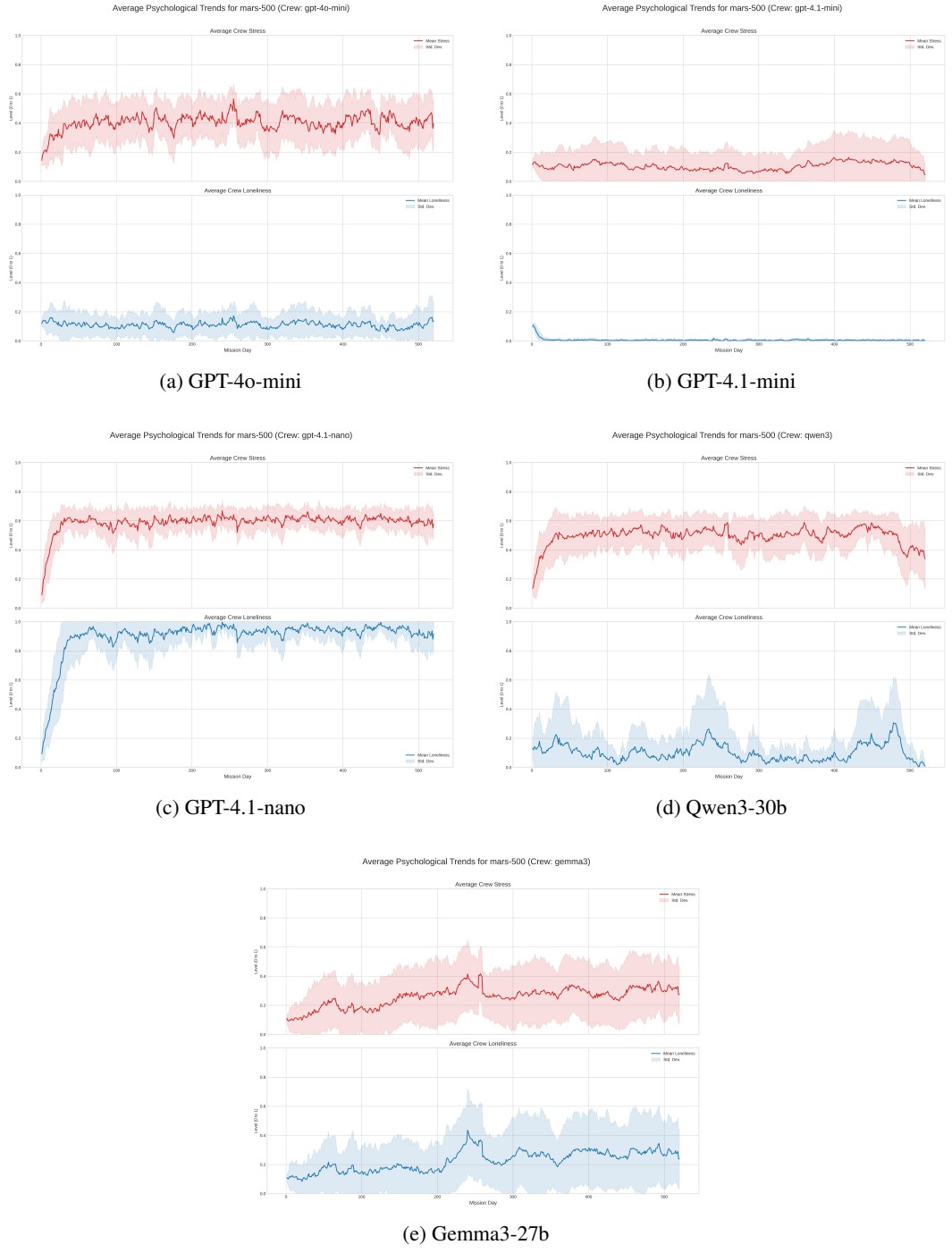

Figure 4: Longitudinal stress dynamics of different models during the Mars-500 simulation **with** the PSA. The intervention is visibly effective at dampening the stress peaks observed in Figure 3, particularly for the more volatile GPT models.


