**Supplementary materials for : PsySpace: Simulating Emergent Psychological Dynamics in Long-Duration Space Missions using Multi-Agent LLMs**

**Reproducibility Statement:**

**Idea Generation:**

For the idea generation. 4 LLMs were prompted to generate the idea after giving initial information, which was a short 2-3 message exchange between the co-authors regarding the broader topic.

**Prompt:**

> Please come up with research Ideas from the above conversation, with a full paper methodology, like method, evaluation , and what to put in the results.
>
> ideas must be novel and for Top AI conferences like NeurIPS. Keep in mind the main topic of research is computational psychology about humans in space.

**Evidence:**

ChatGPT-5.0: https://chatgpt.com/share/68d2051e-21b8-8003-bb2a-089b0d6b452b

Gemini-2.5-Pro: https://g.co/gemini/share/656d12f2a626

DeepSeek R1: See PDF file called "DeepSeek Chat"

Grok-4: https://grok.com/share/c2hhcmQtMg%3D%3D_4b5d478d-b5c6-4d83-a20f-561003b9a495

After obtaining the ideas, the co-authors selected one idea from each model and asked the four models listed above to rank the ideas from 1 to 4. They then chose the idea that received the majority of the votes as the best.

**Evidence:**

ChatGPT-5.0: https://chatgpt.com/share/68d20753-2754-8003-b936-70bddeb14a68

Gemini-2.5-Pro: https://g.co/gemini/share/e6a6fc78c088

DeepSeek R1: See PDF file called "DeepSeek Chat-2"

Grok-4: https://grok.com/share/c2hhcmQtMg%3D%3D_1534b02b-b5ff-4664-a797-ffcd9d404b6e

**Code and Writing:**

For the writing and all the code, the Gemini-2.5-pro model was used. Apart from the Related works, for which we used ChatGPT-5.0's "Deep research" option, we provided it with the generated idea as a Word document.

Related Works (ChatGPT-5.0): https://chatgpt.com/share/68c7eaec-c17c-8008-be9c-9a0bc47f09d2

Code, Writing, and figures (Gemini-2.5-pro): https://g.co/gemini/share/ac867e14a69e

Appendix content (Gemini-2.5-Pro): https://g.co/gemini/share/ab6936a1693d