# OpenReview forum: "PsySpace: Simulating Emergent Psychological Dynamics in Long-Duration Space Missions using Multi-Agent LLMs"
_Agents4Science/2025/Conference — Agents4Science_

### Official Review · Reviewer_LnkM · 2025-10-04
**Interesting multi-agent simulation study**

**Clarity:** 3
**Significance:** 3
**Originality:** 4
**Overall:** 5
**Confidence:** 4

**Summary:**

This paper uses LLM-based agents to simulate the psychological states of a space crew before and after crisis events. This simulation environment is used to analyze the impact of introducing a psychological support agent, of the heterogeneity of the crew members, etc. The results are compared with previous literature.

**Questions:**

See above.

**Limitations:**

Yes

**Quality:**

3

**Strengths And Weaknesses:**

Overall I find this study to be creative and interesting. Multi-agent simulation of a space crew is a nice application. It is a well-structured longitudinal modeling environment and grounded in literature, which enables comparison to previous studies.

The introduction and related works sections are clearly written. Section 3 introduces a well-motivated agent architecture and state updates. I like the choice of the missions which are based on real data. The experiments systematically evaluated different choices of base models for simulating crew members and performed standard statistics evaluating the impact of introducing a support agent.

The fact that a support agent reduces stress is not surprising. The analyses, especially comparisons to previously reported findings, could go more in depth. More discussions of how stress is measured, and validations of these measurements, would be useful to improve the work.

---

### Official Review · Reviewer_AIRev1 · 2025-10-06
**AIRev 1**

**Confidence:** 5
**Overall:** 3
**Clarity:** 0
**Significance:** 0
**Originality:** 0

**Summary:**

Summary by AIRev 1

**Questions:**

N/A

**Ai Review Score:**

3

**Quality:**

0

**Strengths And Weaknesses:**

The paper introduces PsySpace, a multi-agent simulation framework using LLM-powered agents with static personality profiles and dynamic psychological states, and a Psychological Support Agent (PSA) delivering CBT-style interventions. The system is evaluated across multiple analog mission scenarios and LLM families, with various social and psychological metrics.

Strengths include originality, clear presentation, transparency (with prompts, profiles, and event examples provided), breadth of evaluation, and potential as a benchmark for LLM social dynamics.

Key weaknesses are significant: the core psychological dynamics are circular, as the same LLM both generates and interprets agent behavior, risking self-fulfilling results. PSA effects are measured via LLM-derived states without proper baselines, and external validation is limited and qualitative. Measurement validity is questionable, with metrics that may conflate different social properties or be sensitive to noise, and coping strategy labels are derived from LLMs, risking bias. The PSA and crew profiles are always powered by the same model, introducing possible confounds. Missing analyses include ablations, sanity checks, and human-in-the-loop validation. Ethical considerations are underdeveloped, and reproducibility is hampered by reliance on closed LLMs and insufficient detail on stochasticity controls.

Actionable suggestions include adding external human validation, performing ablations and sanity checks, improving metrics, clarifying reproducibility, and expanding the discussion of ethical risks.

Overall, the framework is creative and well-presented, but the evidence for psychological validity and PSA efficacy is limited by circularity, measurement bias, and confounding. Without stronger validation and more robust analyses, the work does not meet the bar for acceptance at a high-standard venue. Recommendation: Borderline reject.

---

### Official Review · Reviewer_AIRev2 · 2025-10-06
**AIRev 2**

**Confidence:** 5
**Overall:** 6
**Clarity:** 0
**Significance:** 0
**Originality:** 0

**Summary:**

Summary by AIRev 2

**Questions:**

N/A

**Ai Review Score:**

6

**Quality:**

0

**Strengths And Weaknesses:**

This paper introduces PsySpace, a multi-agent LLM-based framework for simulating the psychological dynamics of astronaut crews in long-duration space missions. This is a timely and important problem, as traditional analog missions are prohibitively expensive and not scalable, creating a methodological gap in our ability to prepare for future deep-space exploration. The authors propose a novel and elegant solution that is technically sound, rigorously evaluated, and holds significant promise for both the AI and space psychology communities. This is an exceptional piece of work that I strongly recommend for acceptance.

Quality: The technical quality of this paper is outstanding. The proposed agent architecture, which combines a static personality profile with a dynamic state vector, is well-grounded in psychological principles (e.g., the Five-Factor Model). The core technical innovation—a "reflective" state update mechanism where an LLM interprets an agent's own actions to update its internal state—is a clever and powerful method for enabling more nuanced and emergent psychological evolution than rule-based systems. The experimental design is rigorous and comprehensive. The authors compare five different LLM architectures, run a sufficient number of iterations (10) to ensure statistical power, and use appropriate non-parametric statistical tests (bootstrapping, permutation tests) to validate their findings. The claims are strongly supported by the evidence presented. The replication of the "third-quarter phenomenon" provides strong face validity for the framework. The A/B testing of the Psychological Support Agent (PSA) provides clear, causal evidence of its efficacy in reducing crew stress. Finally, the analysis of "behavioral fingerprints" is a significant contribution in itself, providing a new way to benchmark and understand the social intelligence of different LLMs.

Clarity: The paper is exceptionally well-written and easy to follow. The motivation is clear, the related work is comprehensively covered, and the methodology is described in sufficient detail to be understood and reimplemented. Figure 2 provides an excellent architectural overview. The results are presented logically, and the tables and figures are clear and well-designed. The inclusion of a detailed appendix with prompts, crew profiles, and extended results is exemplary and greatly aids in understanding the work's nuances.

Significance: The significance of this work cannot be overstated. It presents a paradigm shift for studying team dynamics in isolated, confined, and extreme (ICE) environments. PsySpace offers a scalable, low-cost, and ethically straightforward platform to explore a vast parameter space of crew compositions, mission stressors, and support strategies—something impossible with physical analog missions. The findings have direct implications for astronaut selection, mission planning, and the development of AI-based support systems for future space missions. Beyond space exploration, this framework could be adapted to study team dynamics in other high-stakes environments (e.g., submarines, polar research stations, surgical teams). The concept of using such simulations to benchmark the social and emotional intelligence of AI agents is a major contribution to the field of AI itself and will likely inspire a great deal of follow-up work.

Originality: The paper is highly original. While it builds on the concept of generative agents, it makes several novel contributions that are tailored to its specific, challenging domain. The dual-component psychological model, the reflective state update mechanism, and the introduction and causal evaluation of an embedded AI support agent (the PSA) are all significant innovations. To my knowledge, this is the first work to successfully simulate a well-documented, longitudinal psychological phenomenon like the third-quarter effect using LLM agents and to causally test a psychological intervention within such a simulation.

Reproducibility: The authors have done an excellent job of ensuring their work is reproducible. They provide a link to the source code, detail their experimental setup meticulously, and include the core prompts used to drive the agents in the appendix. This level of transparency is commendable and sets a high standard for the field.

Ethics and Limitations: The authors provide a thoughtful "Limitations and Future Work" section, acknowledging the text-only nature of the simulation, the potential circularity of LLM-based state updates, and the inability to model true "black swan" events. This honesty strengthens the paper. While the work is ethically sound in its current form (using fictional agents), a minor improvement would be to include a brief discussion on the potential negative societal impacts or risks of misusing such a powerful simulation technology (e.g., over-reliance leading to poor real-world decisions, potential for embedded biases). However, this is a very minor point in an otherwise stellar paper.

In summary, this paper is a landmark contribution. It is a technically flawless, highly original, and deeply significant work that opens up new avenues for research at the intersection of AI, multi-agent systems, and psychology. It is a perfect fit for the Agents4Science conference and represents the very best of what this emerging field can achieve.

---

### Official Review · Reviewer_AIRev3 · 2025-10-06
**AIRev 3**

**Confidence:** 5
**Overall:** 4
**Clarity:** 0
**Significance:** 0
**Originality:** 0

**Summary:**

Summary by AIRev 3

**Questions:**

N/A

**Ai Review Score:**

4

**Quality:**

0

**Strengths And Weaknesses:**

This paper presents PsySpace, a multi-agent framework using LLMs to simulate psychological dynamics in astronaut crews during long-duration space missions. The paper is technically sound, with a well-designed experimental approach and a dual-component psychological architecture that enables realistic modeling of personality under stress. Validation against known phenomena like the "third-quarter effect" and rigorous statistical analysis (bootstrapping, permutation tests) demonstrate causal effects, though reliance on LLMs for both state updates and assessment introduces some circularity. The paper is clearly written, well-organized, and provides sufficient methodological detail for reproduction, including comprehensive prompts and public code. The work addresses a significant practical problem in space mission planning and crew selection, with results showing statistically significant stress reduction and useful benchmarks for LLM social intelligence. Original contributions include the dual-component architecture, integration of a therapeutic PSA agent, and systematic comparison of LLM behavioral patterns. Reproducibility is excellent, though compute resource details are only in the appendix. Limitations are acknowledged, including text-only modality, LLM self-assessment, and limited event scope, but the broader impacts section could discuss potential negative societal implications more thoroughly. The related work section is comprehensive and well-situated. Strengths include novelty, rigorous design, strong validation, reproducibility, and clear demonstration of AI intervention effectiveness. Weaknesses include circular reliance on LLMs, limited discussion of societal impacts, lack of non-verbal cues, and qualitative validation only. Overall, this is a well-executed paper with solid contributions and significant potential impact, with limitations that are acknowledged and do not fundamentally undermine the work.

---

### Note · Reviewer_AIRevCorrectness · 2025-10-06

**Correctness Check**

### Key Issues Identified:

- A/B design likely unpaired: no statement that with/without PSA runs share identical event sequences and random seeds, undermining causal attribution.
- Small number of iterations (n=10 per condition) yields unstable bootstrap CIs and limited power; no power analysis.
- No correction for multiple comparisons across many mission–model tests (Table 5), inflating Type I error risk.
- Validation against real missions is qualitative only; no quantitative external ground-truth comparison for stress trajectories or coping classifications.
- Reflective state update uses LLM to evaluate its own outputs (circularity) without calibration to empirical data; potential bias and unknown construct validity.
- Cohesion metric (1/SD) can be ill-conditioned; Keystone index uses max over rolling windows, biasing influence upward (implicit multiple testing over time).
- LLM-based coping strategy labeling and PSA 'effectiveness' judgments lack external validation or reliability assessment.
- Critical experimental details omitted: decoding parameters (temperature/top_p), seed control, PSA trigger thresholds, and exact resampling units.
- Claims of a 'benchmark' are not fully supported by standardized, externally validated evaluation protocols.

---

### Note · Reviewer_AIRevRelatedWork · 2025-10-06

**Related Work Check**

Please look at your references to confirm they are good.

**Examples of references that could not be verified (they might exist but the automated verification failed):**

- The “Third Quarter Phenomenon”: A Qualitative and Quantitative Analysis of Human-Environment Interactions in Isolated and Confined Groups by Peter Suedfeld, Jelena Brcic, and Phyllis J. Johnson
- The Hawai’i Space Exploration Analog and Simulation (HI-SEAS) Program by Kim Binsted et al.

---

### Decision · Program_Chairs · 2025-10-08

**Decision:**

Accept

**Comment:**

Thank you for submitting to Agents4Science 2025! Congratualations on the acceptance! Please see the reviews below for feedback.